

# Hearing assessment during deep brain stimulation of the central nucleus of the inferior colliculus and dentate cerebellar nucleus in rat

Jasper V. Smit[1,2], Ali Jahanshahi[2,3], Marcus L.F. Janssen[2,4], Robert J. Stokroos[1,2] and Yasin Temel[2,3]

[1] Department of Ear Nose and Throat/Head and Neck Surgery, Maastricht University Medical Center, Maastricht, The Netherlands
[2] Department of Neuroscience, School for Mental Health and Neuroscience, Maastricht University Medical Center, Maastricht, The Netherlands
[3] Department of Neurosurgery, Maastricht University Medical Center, Maastricht, The Netherlands
[4] Department of Neurology, Maastricht University Medical Center, Maastricht, The Netherlands

## ABSTRACT

**Background.** Recently it has been shown in animal studies that deep brain stimulation (DBS) of auditory structures was able to reduce tinnitus-like behavior. However, the question arises whether hearing might be impaired when interfering in auditory-related network loops with DBS.

**Methods.** The auditory brainstem response (ABR) was measured in rats during high frequency stimulation (HFS) and low frequency stimulation (LFS) in the central nucleus of the inferior colliculus (CIC, $n = 5$) or dentate cerebellar nucleus (DCBN, $n = 5$). Besides hearing thresholds using ABR, relative measures of latency and amplitude can be extracted from the ABR. In this study ABR thresholds, interpeak latencies (I–III, III–V, I–V) and V/I amplitude ratio were measured during off-stimulation state and during LFS and HFS.

**Results.** In both the CIC and the CNBN groups, no significant differences were observed for all outcome measures.

**Discussion.** DBS in both the CIC and the CNBN did not have adverse effects on hearing measurements. These findings suggest that DBS does not hamper physiological processing in the auditory circuitry.

Corresponding author
Jasper V. Smit,
jasper.smit@maastrichtuniversity.nl

## INTRODUCTION

Deep brain stimulation (DBS) in auditory structures has been performed in animal studies as a treatment for tinnitus (*Luo et al., 2012*; *Smit et al., 2016*). The rationale behind this treatment is to interfere with the pathological neuronal activity in the central nervous system and interrupt the network loop that is essential for the persistence of tinnitus (*Smit et al., 2015*).

The fundamental knowledge of the effect of deep DBS in auditory structures on hearing is essential before applying this treatment in a clinical setting (*Smit et al., 2015*). It has been shown in rats, using the sound-induced pre-pulse inhibition test with click stimuli, that during high frequency stimulation (HFS) of the external nucleus of the inferior colliculus (IC) hearing thresholds did not change (*Smit et al., 2016*). As far as we know, a more detailed hearing assessment during DBS in auditory structures has not been assessed thus far.

To assess hearing thresholds in more detail, the auditory brainstem response (ABR) was measured in this study. The ABR assesses changes in neural integrity and is commonly used in laboratory animal studies to estimate hearing (*Rosahl et al., 2000*; *Turner et al., 2006*). In humans, ABRs are used in daily practice to assess possible hearing loss of a retrocochlear origin (*Stockard & Rossiter, 1977*).

Two structures were targeted in this study, the central nucleus of the IC (CIC) and the dentate cerebellar nucleus (DCBN). The CIC is the principal auditory part of the IC and has a well-defined tonotopy (*Aitkin & Moore, 1975*; *De Martino et al., 2013*). In animal models of tinnitus, the IC shows tonotopic reorganization, increased spontaneous firing rate, increased bursting activity and increased neural synchrony (*Bauer et al., 2008*; *Chen & Jastreboff, 1995*; *Robertson et al., 2013*; *Wang, Ding & Salvi, 2002*). A recent study showed that HFS of the external nucleus of the IC in rats decreased tinnitus-like behavior (*Smit et al., 2016*). The cerebellum is a structure that is not involved in the auditory pathways but is associated with tinnitus (*Brozoski, Ciobanu & Bauer, 2007*; *Osaki et al., 2005*; *Sedley et al., 2012*; *Shulman & Strashun, 1999*). It was demonstrated that ablation of the paraflocculus completely diminished tinnitus in rats (*Bauer et al., 2013*). The majority of fibers in the cerebellum, including the paraflocculus, originate from the deep cerebellar nuclei, especially the DCBN, which is the largest (*Gayer & Faull, 1988*; *Gould, 1979*). Therefore, the CIC and the DCBN could be considered as respectively an auditory and a non-auditory potential DBS target for the treatment of tinnitus.

DBS can be performed with low frequency stimulation (LFS), which mainly has an excitatory effect, and as HFS, which generally is described as a global inhibitory effect similar as ablation (*Benabid et al., 1998*; *Breit, Schulz & Benabid, 2004*; *Dostrovsky & Lozano, 2002*). Following ablation of IC in animals models, decreased amplitude and latency of peak V have been found (*Achor & Starr, 1980*; *Buchwald & Huang, 1975*; *Durrant et al., 1994*; *Kaga, Shinoda & Suzuki, 1997*). Peak V is the last of the five peaks of the ABR and represents neural activity of the IC. Because of a high variability in amplitude among subjects, the V/I amplitude ratio is a more consistent measure than the absolute value (*Musiek et al., 1984*; *Musiek, Reeves & Baran, 1985*). The relative measures of the latencies are the interpeak latencies (I–III, III–V, I–V), which represent the central transmission latency best (*Eggermont & Don, 1986*; *Picton et al., 1977*; *Squires, Chu & Starr, 1978*). There is little evidence that stimulation of cerebellar structures has influence on the ABR (*Crispino & Bullock, 1984*).

We hypothesized that for CIC stimulation, the V/I amplitude ratio of the ABR would be lower and the I–V or III–V interpeak latencies would be prolonged during HFS and not

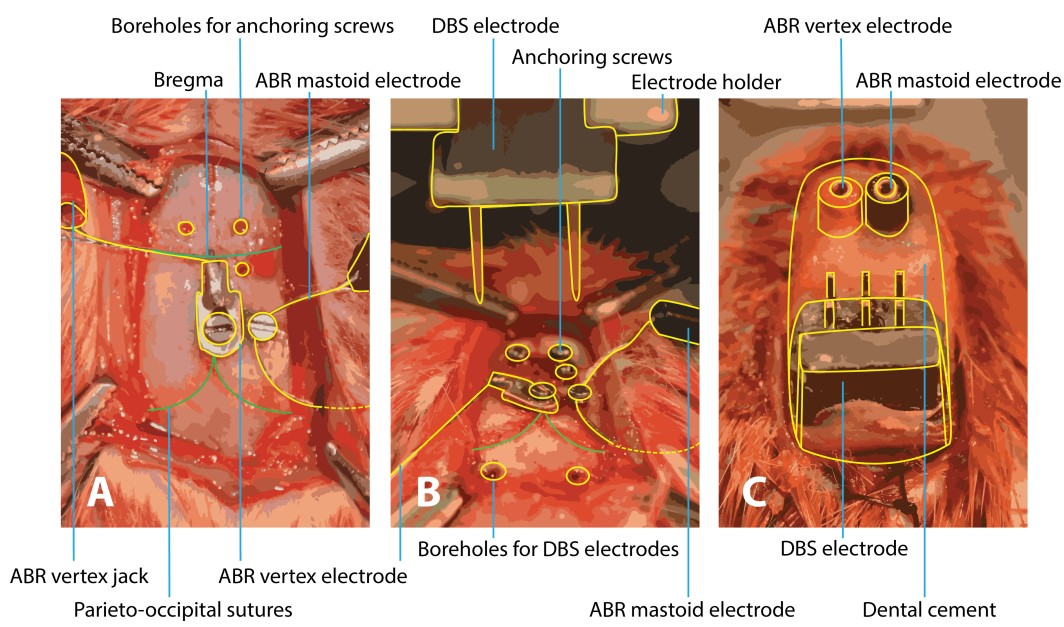

Boreholes for anchoring screws · DBS electrode · ABR vertex electrode
Bregma · ABR mastoid electrode · Anchoring screws · Electrode holder · ABR mastoid electrode
A · B · C
ABR vertex jack · ABR vertex electrode · Boreholes for DBS electrodes · DBS electrode
Parieto-occipital sutures · ABR mastoid electrode · Dental cement

**Figure 1** **Surgery of implantation of ABR and DBS electrodes.** (A) After exposing the skull, the vertex electrode is attached with a screw in the skull and the mastoid electrode is subcutaneously tunneled to the mastoid and also fixated with a screw. Three boreholes are made for anchoring screws to later fixate the structure with dental cement. (B) Boreholes for the DBS electrodes are drilled at coordinates calculated from the bregma level. Calculation of the boreholes and placement of the DBS electrodes are performed within a stereotactic frame. (C) All electrodes are in place and the construct is fixated with dental cement. ABR, auditory brainstem response; DBS, deep brain stimulation.

during LFS of the CIC. Our hypothesis was that stimulating a non-auditory structure such as the DCBN would not have any influence on the ABR.

## METHODS

### Animals

Male rats (Sprague Dawley, 250–300 g; Charles River, Amsterdam, The Netherlands) were housed individually under conditions of constant room temperature and humidity with a reversed 12u/12u light/dark cycle and had *free access* to water and food. The Animal Experiments Committee of the Maastricht University approved the experiments (approval reference number 2012–069).

### Surgical procedure

Subcutaneous electrodes were implanted for ABR recordings and during the same surgery DBS electrodes were implanted in the brain (Fig. 1). Animals were anesthetized by intraperitoneal administration of ketamine (90 mg/kg) and xylazine (10 mg/kg). The head of the rats was immobilized in a stereotactic apparatus (Stoelting Co, Wood Dale, IL, USA) with mouth and blunt ear-bars. Permanent Teflon-coated stainless steel electrodes were subcutaneously implanted. One wire electrode was subcutaneously tunneled to the mastoid and a second wire electrode was attached to a screw on the vertex. Based on coordinates from a stereotactic atlas (*Paxinos & Watson, 2007*), bilateral electrodes (Technomed, Beek,

The Netherlands) were inserted in the CIC (bregma −8.8, depth 4.5, interspace 3.8) or in the DCBN (bregma −11.5, depth 6.5, interspace 6.8). The postoperative recovery time was one week.

## Deep brain stimulation

DBS was performed with bipolar, concentric electrodes using monophasic rectangular pulses. The electrical stimulus pulses were created by an A310 acupulser and an A360 stimulus isolator (World Precision Instruments, Berlin, Germany). During DBS, stimuli were given with a frequency of 100 Hz (HFS) and 10 Hz (LFS) with an amplitude of 100 µA and a pulse width of 60 µs. Electrodes are gold-plated with platinum–iridium inner wire (negative contact) and stainless steel outer part (positive contact). The inner and outer electrodes are insulated except for a 75 µm exposed tip (*Tan et al., 2010*).

Rats were divided in two groups, one group received implantation of electrodes in the CIC ($n = 5$) and the other group in the DCBN ($n = 5$). In the off-stimulation state, designated as the control situation, no electrical stimulation was given. During stimulation-off state, LFS and HFS, ABRs were recorded in separate sessions.

## Auditory brainstem response

ABR measurements were performed in a random manner of the three situations (off-stimulation, LFS, HFS) with a one week interval. Stimulation was turned on approximately 5 min before ABR recordings. HFS consisted of monophasic rectangular pulses, with a frequency of 100 Hz, amplitude of 100 µA per electrode and a pulse width of 60 µs (A310 Acupulser; World Precision Instruments, Berlin, Germany). Similar settings were used in a study which showed tinnitus reduction during HFS in rats (*Smit et al., 2016*). LFS consisted of the same parameters with a frequency of 10 Hz.

To achieve anesthesia during ABR recordings, intraperitoneal administration of ketamine (90 mg/kg) and xylazine (10 mg/kg) was used, which is preferred over isoflurane when assessing hearing thresholds in rats (*Ruebhausen, Brozoski & Bauer, 2012*).

During the ABR procedure, animals were placed into a sound-attenuating chamber. Cables were plugged into the socket of the head of the animal and connected to the recording device (Powerlab 8/35 connected to a Dual Bio Amp amplifier (ADInstruments, Castle Hill, Australia)). An electrode connected to the left hind paw served as the ground.

Custom-made auditory stimuli (10, 16, 24 and 32 kHz) were created with Matlab 2011a (Mathworks, Natick, MA, USA) and consisted of 5 ms bursts with a $\cos^2$ rise and fall filter and were played at a rate of 20 per second at decreasing intensities from 90 to 0 dB peSPL with steps of 10 dB. To prevent synchronous occurrence of stimulation artifacts with the ABRs, one in 10 stimuli had an interval of 55 ms instead of 45 ms. To gain an approximately similar amount of data after filtering of stimulation artifacts, 500 auditory stimuli were given per intensity in the off-stimulation state, 700 during LFS and 1,000 during HFS. Sounds were calibrated with a Bruel & Kjaer 2231 decibel meter with a 4191 microphone (range 2–40 kHz), which was placed at the location of the rat's right ear. Sound intensities are reported as the peak equivalent sound pressure level (peSPL).

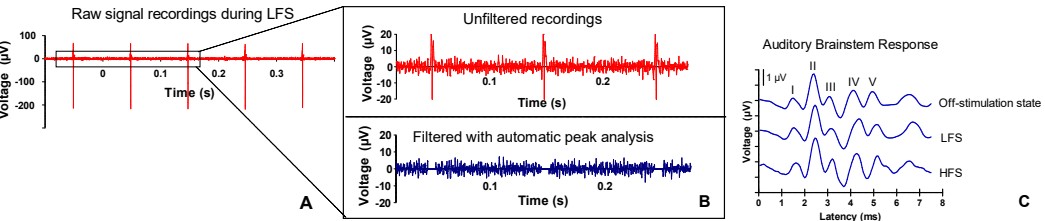

**Figure 2** **ABR signal processing.** (A) Example of a raw signal that was measured during low frequency stimulation (LFS). (B) Stimulation artifacts are filtered with automatic peak detection analysis. (C) Example of an auditory brainstem response (ABR) (burst frequency 10 kHz) during off-stimulation state, during LFS and during high frequency stimulation (HFS) in the central nucleus of the inferior colliculus. The five ABR peaks are numbered I–V. Morphology and latency of ABR peaks in the current study were consistent with other animal studies (*Backoff & Caspary, 1994*; *Dehmel, Eisinger & Shore, 2012*; *Zheng et al., 2012*). The first peak arises approximately 1.5 ms after stimulus onset. Although there is overlap, the first peak represents neural activity of the cochlear nerve. The second peak is considered to be mainly generated by cochlear nuclear cells, the third peak by the contralateral superior olivary complex cells and the fourth peak by the lateral lemniscus. The fifth peak originates from the inferior colliculus (*Biacabe et al, 2001*; *Chen & Chen, 1991*; *Simpson et al., 1985*).

Auditory stimuli were processed with an external soundcard with a sample rate of 192 kHz (Creative E-MU 0204), amplified with Ultrasonic power amplifier (Avisoft Bioacoustics, Berlin) and played with an Ultrasonic Dynamic Speaker Vifa (Avisoft Bioacoustics, Berlin, Germany) to the right ear. To standardize sound presentation between recording sessions it was monitored that in every session the same position of the rat and the same distance between the loudspeaker and the ear was used (2 cm). The contralateral ear was plugged with modeling clay.

Auditory stimuli were digitally triggered. The recordings were done in Labchart Pro 7 (ADInstruments, Castle Hill, Australia) at a sample frequency of 20 kHz and raw data were imported into Matlab. With a customized script, the signal was amplified 100,000 times and band-pass filtered (300–3,000 Hz). Evoked responses were averaged and data which contained DBS artifacts were automatically removed based on a peak-detection analysis. Using a customized Matlab script, peaks were automatically detected if the signal was above a manual depicted maximal baseline value. Before and after the maximal value of the peak of the artifact 2.5 ms of data were converted in Not-a-Number (NaN). The ABR and DBS stimuli were not phase-locked so per epoch a different part was converted in NaN. All epochs were averaged to calculate the mean ABR signal (Fig. 2B).

Two independent blinded observers visually identified ABR thresholds and peaks. In case of disagreement, a third observer was sought and the concordant data were accepted. The auditory threshold was defined as the lowest decibel level (peSPL) of the stimulus, which produced a distinctive ABR.

For latency analysis, the five positive peaks were determined at 90 dB peSPL and numbered I–V based on the recordings of vertex upward deflections (for an example see Fig. 2C). Latencies of peaks were measured from stimulus onset. Interpeak latency was defined as the time between respective peaks.

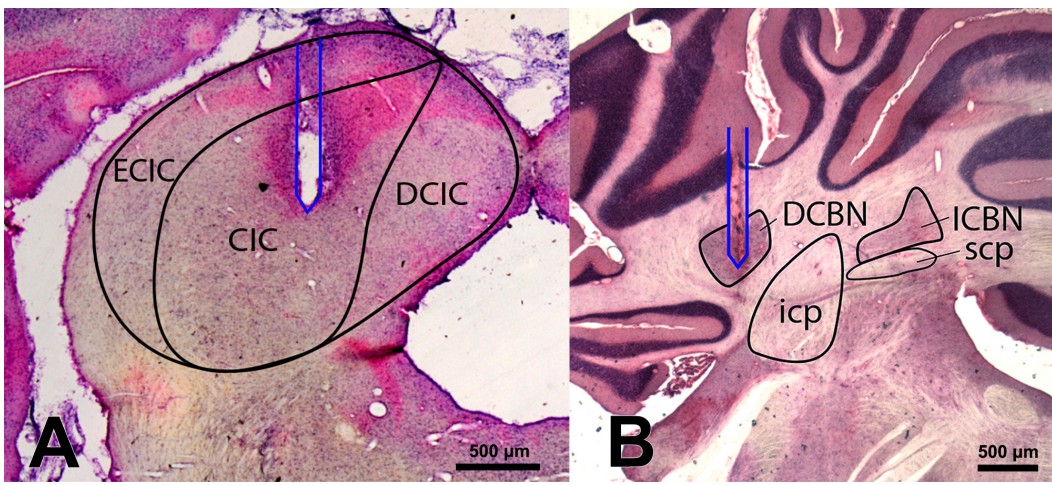

**Figure 3 Histology.** Representative examples of electrode positions (white lines) in the CIC (A) and DCBN (B). All electrodes were implanted bilaterally. ECIC, external nucleus of the inferior colliculus; CIC, central nucleus of inferior colliculus; DCIC, dorsal cortex of inferior colliculus; DCBN, dentate cerebellar nucleus; icp, inferior cerebellar peduncle; ICBN, interposed cerebellar nucleus; scp, superior cerebellar peduncle. Scale bar: 500 μm.

The amplitude was expressed as the peak-to-peak amplitude ratio of peak V subtracted by peak I.

## Electrode localization

Animals were deeply anesthetized with pentobarbital (75 mg/kg) and perfused transcardially with Tyrode's buffer (0.1 M) and fixative containing 4% paraformaldehyde, 15% picric acid and 0.05% glutaraldehyde in 0.1 M phosphate buffer (pH 7.6). After post-fixation for 12 h, the brains were cut to coronal sections using a vibrotome. To assess the electrode localization, the sections containing the target area and the electrode trajectory were stained with hematoxylin-eosin (Merck, Darmstadt, Germany). Definition of anatomic structures was based on the stereotactical atlas (*Paxinos & Watson, 2007*).

## Statistical analysis

Dependent data were analyzed using the Wilcoxon signed-rank Test for two groups and a Friedman test for multiple groups. Since multiple comparisons were made when comparing the stimulation-off state with LFS and HFS, modified $p$-values (alpha = 0.05) are given as corrected by means of the Holm-Bonferonni sequential correction (*Holm, 1979*). Data are presented as mean ± standard error of the mean (SEM). All data were analyzed with SPSS (Version 20, IBM, Somers, NY, USA).

## RESULTS

### Electrode localization

Histological evaluation showed that all electrodes were implanted correctly in the target structures (Fig. 3A and Fig. 3B).

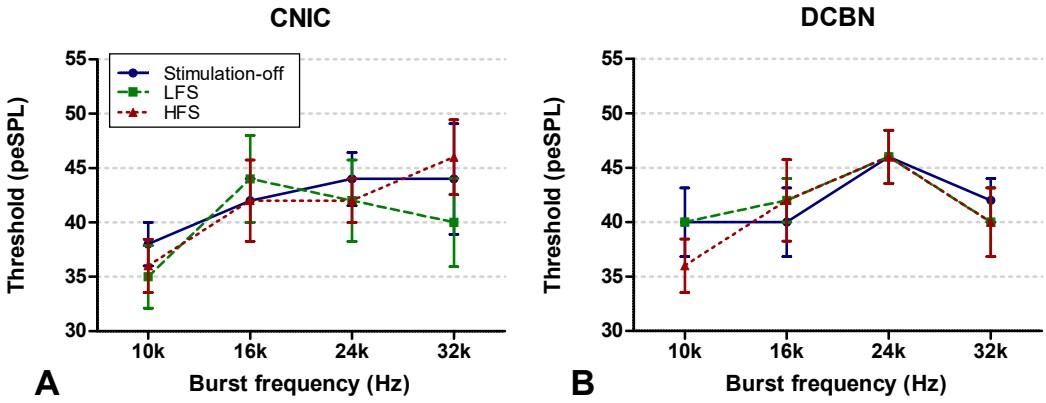

**Figure 4 ABR thresholds.** ABR thresholds of the CIC (A) and DCBN group (B) measured during the DBS-off state (blue, circles, solid line), LFS (green, squares, striped line) and HFS (red, triangles, dotted line). There was no statistically significant difference. The vertical lines indicate the standard error of the mean. ABR, auditory brainstem response; CIC, central nucleus of the inferior colliculus; DBS, deep brain stimulation; DCBN, dentate cerebellar nucleus; LFS, low frequency stimulation; HFS, high frequency stimulation.

### Hearing thresholds

Hearing response thresholds were determined as the minimal intensity stimulus at which an ABR was evident. Thresholds of different stimulus frequencies (10, 16, 24 and 32 kHz) are depicted in Fig. 4A for the CIC group and in Fig. 4B for the DCBN group. In one rat two thresholds (10 Hz LFS and 32 Hz LFS) were not possible to determine. In both groups, no statistically significant differences were found during HFS and LFS compared to off-stimulation.

### Latencies and amplitudes

From all ABRs, 5 distinctive peaks could be determined at 90 dB peSPL (Fig. 1). In Table 1 the mean interpeak latencies (I–III, III–V and I–V) are shown for different burst frequencies (10, 16, 24 and 32 kHz). In both the CIC and the DCBN group, no statistically significant differences were found for high and low frequency DBS compared to no stimulation (Table 1).

The V/I amplitude ratio was calculated at all burst frequencies. In both groups, there was no statistical significant difference when comparing no stimulation with HFS and LFS. Tables A1–A4 in Appendix shows the absolute latencies and interpeak latencies.

When looking at the latency and amplitude data, a relation between ABR latencies and amplitudes, with frequencies of burst tones was noticed. For further analysis, we grouped the off-stimulation data of the CIC and DCBN group since only baseline measurements were analyzed. The latency, e.g., of peak I, differed between burst frequencies ($X^2(3) = 20.12$, $p < 0.01$). The raw data (see Appendix) show a shorter latency with increasing frequencies of burst tones. The V/I amplitude ratio does not differ amongst frequencies ($X^2(3) = 4.92$, $p = .178$). Amplitudes of peak I did not differ between frequencies ($X^2(3) = 3.240$, $p = .355$), but the amplitude of peak V was different between frequencies ($X^2(3) = 17.160$, $p < 0.01$). Also peak V amplitude decreases with increasing burst frequency.

Smit et al. (2017), *PeerJ*, DOI 10.7717/peerj.3892

**Table 1 Interpeak latencies (IL) and V/I amplitude ratio (AR) for 10 k, 16 k, 24 k and 32 k burst sounds.** Mean values with standard deviation are given. Adjusted Holm-Bonferroni pvalues are used.

| Frequency (Hz) | Peaks | CIC group | | | | | DCBN group | | | | |
|---|---|---|---|---|---|---|---|---|---|---|---|
| | | Stim-off | LFS | p | HFS | p | Stim-off | LFS | p | HFS | p |
| 10 k | IL I–III | 1.504 (.170) | 1.524 (.125) | >.99 | 1.470 (.285) | >.99 | 1.520 (.157) | 1.590 (.162) | 0.20 | 1.530 (.136) | >.99 |
| | IL III–V | 1.908 (.213) | 1.925 (.128) | >.99 | 1.745 (.285) | >.99 | 1.842 (.028) | 1.842 (.116) | >.99 | 1.953 (.109) | .32 |
| | IL I–V | 3.411 (.312) | 3.449 (.143) | >.99 | 3.24 (.291) | >.99 | 3.362 (.153) | 3.433 (.245) | >.99 | 3.514 (.130) | .22 |
| | AR V/I | 1.542 (1.739) | 1.238 (.832) | >.99 | 1.174 (.517) | >.99 | 1.349 (.946) | 1.1310 (.398) | >.99 | 1.326 (.287) | >.99 |
| 16 k | IL I–III | 1.560 (.160) | 1.540 (.131) | >.99 | 1.490 (.145) | 0.25 | 1.560 (.147) | 1.540 (.084) | >.99 | 1.561 (.062) | >.99 |
| | IL III–V | 1.822 (.257) | 1.943 (.091) | >.99 | 1.862 (.113) | >.99 | 1.863 (.113) | 1.933 (.166) | .85 | 1.9630 (.050) | >.99 |
| | IL I–V | 3.383 (.395) | 3.483 (.218) | >.99 | 3.353 (.248) | >.99 | 3.423 (.252) | 3.474 (.214) | >.99 | 3.5236 (.094) | >.99 |
| | AR V/I | .979 (.410) | .9067 (.320) | .69 | 1.310 (.410) | >.99 | 1.178 (.913) | 1.278 (1.470) | >.99 | 1.078 (.289) | >.99 |
| 24 k | IL I–III | 1.550 (.157) | 1.550 (.130) | >.99 | 1.570 (.232) | >.99 | 1.560 (.113) | 1.520 (.090) | >.99 | 1.621 (.120) | >.99 |
| | IL III–V | 1.853 (.186) | 2.004 (.135) | 0.25 | 1.974 (.180) | >.99 | 1.712 (.202) | 1.893 (.293) | .25 | 1.750 (.200) | >.99 |
| | IL I–V | 3.403 (.329) | 3.554 (.243) | 0.26 | 3.544 (.309) | >.99 | 3.272 (.250) | 3.413 (.279) | >.99 | 3.371 (.147) | .51 |
| | AR V/I | .957 (.242) | .916 (.594) | >.99 | 1.11 (.595) | >.99 | .536 (.265) | .445 (.305) | .50 | .834 (.470) | .50 |
| 32 k | IL I–III | 1.550 (.109) | 1.478 (.093) | >.99 | 1.428 (.261) | >.99 | 1.570 (.097) | 1.520 (.066) | .23 | 1.570 (.157) | >.99 |
| | IL III–V | 1.883 (.318) | 2.010 (.376) | >.99 | 1.850 (.240) | >.99 | 1.822 (.065) | 1.903 (.272) | >.99 | 1.862 (.366) | >.99 |
| | IL I–V | 3.433 (.400) | 3.488 (.372) | >.99 | 3.278 (.416) | >.99 | 3.393 (.145) | 3.423 (.283) | >.99 | 3.433 (.362) | >.99 |
| | AR V/I | .491 (.437) | .505 (.278) | .69 | .617 (.523) | .28 | .404 (.296) | .551 (.209) | .50 | .870 (.961) | .45 |

**Notes.**

Abbreviations: CIC, central nucleus of interior colliculus; DCBN, dentate cerebellar nucleus; stim-off, stimulation-off state; LFS, deep brain stimulation at 10 Hz; HFS, deep brain stimulation at 100 Hz.

## DISCUSSION

We successfully measured ABRs during stimulation-off state, LFS and HFS. Our results showed that LFS as well as HFS in the CIC and DCBN do not influence ABR thresholds, interpeak latencies and amplitude ratios in rats.

### ABR thresholds

The finding that ABR thresholds were not influenced by LFS and HFS suggests that hearing in these frequencies is not impaired by DBS. Nonetheless, several caveats must be taken into account when interpreting ABR thresholds. Although common frequencies were tested (10, 16, 24 and 32 kHz) in these studies, hearing loss can occur in other specific frequency bands. In rats, hearing thresholds based on ABRs tend to be at least 10–20 dB higher than those determined behaviorally (*Borg, 1982*; *Heffner et al., 1994*). The thresholds in the current study (ranging from 36 to 46 dB peSPL) are thus an overestimation of the actual hearing level. To get the most reproducible ABR data in various measurements, we implanted ABR electrodes. In contrast to the commonly used subcutaneous electrodes, these implanted electrodes always measure from exactly the same anatomical position (*Buchwald et al., 1981*; *Hall, 1990*; *McGee, Ozdamar & Kraus, 1983*). To our knowledge, no other studies determined ABR thresholds during HFS and LFS of the CIC or DCBN. Likewise, determination of thresholds in ablation studies, whose results are thought to be similar to HFS, have not been performed.

### ABR latency

In addition to thresholds, the latency and amplitude can be extracted from the five ABR peaks. Interpeak latencies are generally accepted as measures of conduction time of the central auditory pathway (*Eggermont & Don, 1986*; *Picton et al., 1977*; *Squires, Chu & Starr, 1978*). The interpeak latency of waves I–III, III–V and I–V reflect the time to traverse in the caudal, rostral and the whole brainstem, respectively. A prolonged interpeak latency reflects a lesion in central auditory processing (*Burkhard, Eggermont & Don, 2007*; *Hood, 1998*). Occasionally, a decreased latency of peak V was noted in ablation studies of the IC. This decrease of peak V latency was only an acute effect (*Achor & Starr, 1980*).

In this study, no statistically significant differences were found between the interpeak latencies at baseline compared to low and high frequency DBS. This can be interpreted as no functional relevant lesion at the IC is induced by DBS. However, many studies found no differences in latencies when ablating the IC, but found a difference in amplitude (*Achor & Starr, 1980*; *Buchwald & Huang, 1975*; *Caird & Klinke, 1987*). Therefore, we also performed analysis of the ABR amplitude.

### ABR amplitude

Synchronously activated neurons contribute to the amplitude of the waveform (*Burkhard, Eggermont & Don, 2007*). The IC has a central role in the auditory pathway (*Aitkin & Moore, 1975*; *De Martino et al., 2013*). Previous studies have shown that lesioning of the IC resulted in a decrease of the amplitude of peak V (*Achor & Starr, 1980*; *Buchwald & Huang, 1975*; *Caird & Klinke, 1987*). In most studies a large part or the whole IC was ablated. One

study only found an abolished peak V when ablation of the lateroventral part of the IC, in contrast to ablating the central nucleus (*Funai & Funasaka, 1983*). In humans, absence of the IC also resulted in abolished peak V peaks (*Durrant et al., 1994*). It is assumed that electrode implantation does not influence the amplitude of the evoked potentials, since only minimal tissue damage is seen along the electrode trajectory (*Tan et al., 2010*).

Although the precise role of the cerebellum and its associated nuclei in hearing is not known, it might have a modulatory effect on hearing. The cerebellum receives direct connections from the cochlear nucleus (*Huang, Liu & Huang, 1982*) and indirect connections from the IC (*Aitkin & Boyd, 1978*; *Huffman & Henson, 1990*). Furthermore, auditory stimuli as well as stimulation of the auditory cortex elicited responses from auditory cells in the paraflocculus (*Azizi, Burne & Woodward, 1985*).

One study assessed the ABR during cerebellar stimulation. High frequency stimulation (400 Hz) of the cerebellar surface resulted in a difference of the IV/I amplitude ratio, where peak IV represented in this particular study the IC. The IV/I amplitude ratio increased in case of a short electrical-sound stimulus interval (<10 ms), and decreased with larger intervals (>10 ms). In this particular study, peak IV represented the IC (*Crispino & Bullock, 1984*). In our study, the electrical and sound stimuli were played in an asynchronous manner and therefore various interval times are achieved. This could explain why we did not found any difference in the amplitude ratio. As far as we know, no ABRs were recorded in a cerebellar ablation study.

## General ABR findings

It is a well-known phenomenon that high frequency tones show shorter latency peaks than lower frequency sounds, because high frequency sounds stimulate the more basal portions of the basilar membrane (*Alvarado et al., 2012*). This is also seen in our data. We also found that the peak V amplitude ratio decreased with increasing frequency of the tone given. As far as we know this is a new finding, which has not been reported earlier.

## Mechanism of DBS in the auditory system

Our results show that latencies were not prolonged and amplitudes were not decreased during DBS, indicating that DBS in the CIC and DCBN probably does not have an overall inhibitory effect on physiological central auditory processing up to the IC (peak V). This finding is supported by one of the main working mechanisms of DBS. Namely that DBS with frequencies above 100 Hz disrupts abnormal information flow in a network (*Chiken & Nambu, 2014*), without influencing the normal neurophysiological activity.

HFS is also often referred to as having an inhibitory effect and thus mimicking the effect of a lesion (*Benabid et al., 1998*; *Dostrovsky & Lozano, 2002*). The pathological neural network loop related to tinnitus is interrupted by performing HFS within this loop (*Smit et al., 2016*). This hypothesis is supported by the disruption theory; DBS can dissociate the input and output in a stimulation nucleus and thereby disrupting abnormal information flow such as increased burst activity. Physiological information can still be normally processed through different nuclei (*Chiken & Nambu, 2014*). It can be hypothesized that this is the same when DBS is applied in the auditory pathway and physiological auditory information processing remains intact.

## Future studies

In the current study animal did not receive noise trauma for induction of tinnitus. We hypothesize that if DBS does not result in hearing loss in the normal hearing, this will also not be the case when there is hearing loss in association with tinnitus. The current stimulation parameters can be used for tinnitus treatment; in a recent study that showed a decrease of tinnitus during IC stimulation (*Smit et al., 2016*), the same stimulation parameters were used as in the current study. In our study no pre-operative assessment of the ABR was performed.

## Conclusions

In conclusion, HFS and LFS in the CIC and DCBN did not result in increased ABR thresholds and changes in interpeak latencies. Based on these observations, no evidence for changes in information processing in the auditory circuit were found during low and high frequency DBS in the CIC and DCBN. These findings suggest that DBS in the auditory pathways can be performed without hampering physiological processing of auditory information.

## APPENDIX: ABSOLUTE VALUES OF LATENCIES AND AMPLITUDES

**Table A1 Absolute values of latencies and amplitudes from the five peaks of the auditory brainstem response of 10 kHz auditory stimuli.** Mean values with standard deviation are given.

| Peak | Wave | CIC group | | | DCBN group | | |
|---|---|---|---|---|---|---|---|
| | | Stim-off | LFS | HFS | Stim-off | LFS | HFS |
| Latencies | I | 1.639 (.073) | 1.599 (.110) | 1.618 (.080) | 1.540 (.165) | 1.490 (.076) | 1.570 (.090) |
| | II | 2.460 (.092) | 2.479 (.121) | 2.392 (.079) | 2.386 (.162) | 2.325 (.066) | 2.356 (.083) |
| | III | 3.143 (.181) | 3.122 (.208) | 3.088 (.144) | 3.060 (.280) | 3.080 (.109) | 3.101 (.084) |
| | IV | 4.288 (.281) | 4.167 (.191) | 4.122 (.238) | 4.077 (.260) | 4.118 (.165) | 4.148 (.098) |
| | V | 5.050 (.275) | 5.047 (.180) | 4.765 (.291) | 4.903 (.270) | 4.923 (.206) | 4.983 (.015) |
| Amplitudes | I | .007 (.018) | .028 (.030) | .037 (.042) | .030 (.016) | .030 (.011) | .041 (.047) |
| | II | .221 (.018) | .218 (.042) | .252 (.054) | .159 (.012) | .181 (.009) | .185 (.048) |
| | III | .042 (.018) | .074 (.037) | .076 (.038) | .078 (.031) | .097 (.066) | .117 (.069) |
| | IV | .070 (.017) | .086 (.042) | .117 (.035) | .061 (.055) | .087 (.041) | .103 (.053) |
| | V | .030 (.059) | .054 (.049) | .063 (.043) | .077 (.028) | .069 (.023) | .093 (.039) |

Notes.

Abbreviations: CIC, central nucleus of interior colliculus; DCBN, dentate cerebellar nucleus; stim-off, stimulation-off state; LFS, deep brain stimulation at 10 Hz; HFS, deep brain stimulation at 100 Hz.

Table A2  **Absolute values of latencies and amplitudes from the five peaks of the auditory brainstem response of 16 kHz auditory stimuli.** Mean values with standard deviation are given.

| Peak | Wave | CIC group | | | DCBN group | | |
|------|------|-----------|---|---|------------|---|---|
| | | Stim-off | LFS | HFS | Stim-off | LFS | HFS |
| Latencies | I | 1.400 (.153) | 1.419 (.042) | 1.530 (.076) | 1.399 (.042) | 1.389 (.027) | 1.470 (.066) |
| | II | 2.245 (.153) | 2.275 (.055) | 2.325 (.075) | 2.215 (.128) | 2.285 (.084) | 2.275 (.065) |
| | III | 2.960 (.292) | 2.960 (.125) | 3.020 (.155) | 2.96 (.140) | 2.929 (.075) | 3.030 (.090) |
| | IV | 3.956 (.367) | 4.067 (.190) | 4.057 (.230) | 3.977 (.118) | 4.027 (.155) | 4.098 (.145) |
| | V | 4.782 (.534) | 4.903 (.206) | 4.883 (.249) | 4.822 (.240) | 4.863 (.213) | 4.993 (.120) |
| Amplitudes | I | .042 (.040) | .039 (.031) | .037 (.027) | .036 (.018) | .025 (.016) | .058 (.046) |
| | II | .182 (.028) | .162 (.030) | .189 (.043) | .139 (.052) | .119 (.018) | .178 (.023) |
| | III | .079 (.048) | .080 (.048) | .085 (.029) | .088 (.047) | .079 (.040) | .118 (.052) |
| | IV | .050 (.019) | .057 (.031) | .089 (.029) | .047 (.015) | .072 (.030) | .113 (.035) |
| | V | .067 (.047) | .050 (.028) | .077 (.029) | .080 (.089) | .034 (.023) | .086 (.040) |

**Notes.**
Abbreviations: CIC, central nucleus of interior colliculus; DCBN, dentate cerebellar nucleus; stim-off, stimulation-off state; LFS, deep brain stimulation at 10 Hz; HFS, deep brain stimulation at 100 Hz.

Table A3  **Absolute values of latencies and amplitudes from the five peaks of the auditory brainstem response of 24 kHz auditory stimuli.** Mean values with standard deviation are given.

| Peak | Wave | CIC group | | | DCBN group | | |
|------|------|-----------|---|---|------------|---|---|
| | | Stim-off | LFS | HFS | Stim-off | LFS | HFS |
| Latencies | I | 1.379 (.126) | 1.379 (.027) | 1.480 (.131) | 1.299 (.083) | 1.399 (.075) | 1.419 (.066) |
| | II | 2.235 (.200) | 2.215 (.071) | 2.293 (.111) | 2.134 (.116) | 2.225 (.083) | 2.305 (.066) |
| | III | 2.929 (.234) | 2.929 (.114) | 3.050 (.131) | 2.859 (.140) | 2.919 (.094) | 3.040 (.121) |
| | IV | 3.896 (.326) | 3.987 (.186) | 4.097 (.272) | 3.906 (.170) | 4.007 (.116) | 4.108 (.166) |
| | V | 4.782 (.391) | 4.933 (.228) | 5.023 (.262) | 4.570 (.215) | 4.812 (.275) | 4.790 (.129) |
| Amplitudes | I | .026 (.025) | .023 (.038) | .033 (.027) | .032 (.011) | .028 (.014) | .046 (.045) |
| | II | .116 (.020) | .094 (.020) | .110 (.028) | .090 (.016) | .088 (.015) | .117 (.029) |
| | III | .080 (.062) | .072 (.062) | .074 (.031) | .070 (.033) | .055 (.034) | .085 (.072) |
| | IV | .037 (.033) | .192 (.301) | .083 (.054) | .053 (.024) | .054 (.009) | .077 (.041) |
| | V | .034 (.036) | .033 (.045) | .043 (.028) | .022 (.014) | .013 (.013) | .050 (.039) |

**Notes.**
Abbreviations: CIC, central nucleus of interior colliculus; DCBN, dentate cerebellar nucleus; stim-off, stimulation-off state; LFS, deep brain stimulation at 10 Hz; HFS, deep brain stimulation at 100 Hz.

**Table A4  Absolute values of latencies and amplitudes from the five peaks of the auditory brainstem response of 32 kHz auditory stimuli.** Mean values with standard deviation are given.

| Peak | Wave | CIC group | | | DCBN group | | |
|------|------|-----------|-----|-----|-----------|-----|-----|
| | | Stim-off | LFS | HFS | Stim-off | LFS | HFS |
| Latencies | I | 1.369 (.826) | 1.408 (.051) | 1.497 (.188) | 1.289 (.104) | 1.268 (.042) | 1.409 (.155) |
| | II | 2.235 (.145) | 2.222 (.106) | 2.272 (.122) | 2.104 (.180) | 1.980 (.444) | 2.325 (.153) |
| | III | 2.919 (.155) | 2.885 (.124) | 2.926 (.139) | 2.859 (1.80) | 2.789 (.098) | 2.980 (.199) |
| | IV | 3.946 (.210) | 3.912 (.130) | 3.982 (.207) | 3.866 (.232) | 4.097 (.614) | 4.027 (.216) |
| | V | 4.802 (.413) | 4.896 (.371) | 4.776 (.304) | 4.681 (.222) | 4.691 (.271) | 4.842 (.277) |
| Amplitudes | I | .017 (.011) | .024 (.032) | .052 (.124) | .035 (.017) | .023 (.013) | .043 (.041) |
| | II | .097 (.019) | .156 (.150) | .167 (.225) | .078 (.021) | .091 (.030) | .092 (.030) |
| | III | .062 (.015) | .093 (.091) | .134 (.145) | .058 (.031) | .056 (.063) | .058 (.040) |
| | IV | .057 (.048) | .129 (.141) | .190 (.268) | .034 (.026) | .048 (.028) | .066 (.042) |
| | V | .021 (.022) | .032 (.054) | .039 (.090) | .015 (.146) | .013 (.025) | 0.042 (.043) |

**Notes.**

Abbreviations: CIC, central nucleus of interior colliculus; DCBN, dentate cerebellar nucleus; stim-off, stimulation-off state; LFS, deep brain stimulation at 10 Hz; HFS, deep brain stimulation at 100 Hz.

### Funding

This study was supported by the Heinsius Houbolt Foundation. The funders had no role in study design, data collection and analysis, decision to publish, or preparation of the manuscript.

### Grant Disclosures

The following grant information was disclosed by the authors:
Heinsius Houbolt Foundation.

### Competing Interests

The authors declare there are no competing interests.

### Author Contributions

- Jasper V. Smit conceived and designed the experiments, performed the experiments, analyzed the data, contributed reagents/materials/analysis tools, wrote the paper, prepared figures and/or tables, reviewed drafts of the paper.
- Ali Jahanshahi and Marcus L.F. Janssen conceived and designed the experiments, performed the experiments, analyzed the data, contributed reagents/materials/analysis tools, wrote the paper, reviewed drafts of the paper.
- Robert J. Stokroos and Yasin Temel conceived and designed the experiments, wrote the paper, reviewed drafts of the paper.

### Animal Ethics

The following information was supplied relating to ethical approvals (i.e., approving body and any reference numbers):

The Animal Experiments Committee of Maastricht University approved the experiments.

## Data Availability

The raw data has been supplied as a Data S1.

## Supplemental Information

Supplemental information for this article can be found online at http://dx.doi.org/10.7717/peerj.3892#supplemental-information.

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
