# Peer review of "Hearing assessment during deep brain stimulation of the central nucleus of the inferior colliculus and dentate cerebellar nucleus in rat"

_PeerJ, doi:10.7717/peerj.3892_

## Round 0.1 · original submission · Major Revisions

Dear Authors,

There are major comments from the first peer reviewer that will be important for your team to revise and improve the presentation of the manuscript.

Thanking You.

·

Basic reporting

Entirely adequate.

Experimental design

Methods organization could be improved.
The section describing DBS was buried in the ABR method section. It should have been a separate and more complete section, as it was in the Smit et al., (2016) research report.
The low n (5 / group) of the present study sets it up for failure to find significant effects. This appears particularly true since non-parametric statistical tests, which do not take advantage of the repeated-measures design, were used. The authors should explain their choice of statistical tests and consider the issue of adequate statistical power to determine significance.
Although it wasn’t well explained (but should have been), apparently DBS was delivered throughout the epoch devoted to ABR acquisition. Hence there was a need to remove stimulus artifacts, as shown, but not thoroughly explained, in Figure 2. Apparently the DBS, both high-frequency and low, were delivered without controlling phase relationship to the ABR acquisition. That experimental choice should be justified. DBS periods were either 10 msec or 100 msec. ABR inter-stimulus intervals were 45 msec, while the ABR recording periods were 50 msec. Rather than let these events wander in and out of phase, a more tightly controlled protocol could have been used. For example, if the ABR-eliciting sound pulses were out of phase with DBS, then the DBS stimulus artifact would not have to be filtered. This would eliminate the loss of data produced by EP filtering.

Validity of the findings

Perhaps the most important results appear in Figure 4. In my opinion it would have been more informative to show thresholds, or ABR signal levels (such as RMS), separately for waves I through V. This, because it is surprising that DBS in the IC would not affect ABR wave V amplitudes, since the IC is a major source of wave V potentials. It wasn’t clear to me how the wave V/I amplitude ratios, reported in Table 1, addressed the issue of amplitude effects. Were all of the Table 1 data collected with sound pulses at 90 dB (SPL)? This was implied in Methods, but not clear. If yes, what about the dynamic range of wave V (and other wave components)? Furthermore, if both wave I and wave V diminished, the V/I ratio would not change.
On the topic of threshold: Threshold is just one point of the dynamic range of evoked potentials. In this context Schaette (Schaette, 2014) and others have introduce the concept of “hidden hearing loss.” A more sensitive measure of loss would be to quantify the evoked response strength over the entire range of stimulus levels, for each test frequency and/or each ABR wave (I – V). Root mean square (RMS) is often used for that purpose. Another advantage of the dynamic-range approach is that thresholds do not have to be determined by experimenter judgment.
One level of DBS (100 µA) was tested. Since this level was shown in the preceding tinnitus study (Smit, et al., 2016) to be effective in tinnitus attenuation, perhaps this is adequate to show that the tinnitus attenuation did not derive from a hearing threshold shift.. However, the value of the present study would have been improved if a range of DBS levels had been tested… analogous to testing a drug at more than one dose.

Additional comments

The primary value of the present research might be the addition of the DCBN component. Note the published experiment (Smit et al., 2016) showed that DBS in the IC did not affect PPI. On this basis one would have to conclude that IC DBS did not interfere with central auditory processing. This would be true because PPI depends not only upon a functioning auditory brainstem (and ear), but also to some extent, a functioning auditory forebrain (Du et al., 2011). The point is that the prior published finding diminishes the knowledge added by the present work. Nevertheless, demonstrating an effect, or lack of effect, on the ABR has value.
References
Du, Y., Wu, X., Li, L. 2011. Differentially organized top-down modulation of prepulse inhibition of startle. J Neurosci 31, 13644-53.
Schaette, R. 2014. Tinnitus in men, mice (as well as other rodents), and machines. Hear Res 311, 63-71.

·

Basic reporting

The article is written in clear English and is well structured. The hypotheses are clearly explained, reasonable and supported by sufficient amount of existing work in the literature. The structure of the result reflects what is expected from the hypotheses tested by the authors.

Experimental design

In this article, the authors test the potential of deep brain stimulation to treat tinnitus in two brain targets. Both targets are theoretically relevant and untested, moreover, they chose to implant the electrodes in the area, which is involved directly in auditory input processing and another area, which is not involved, but remains relevant based on lesion studies.
The scientific is well formulated and directly tested.
The investigation was performed rigorously and the authors have considerable experience with deep brain stimulation experimental design. This particularly visible in the method description section, in which is all relevant experimental details are well explained to allow for data reproduction.

Validity of the findings

Although the data are mostly not significant, the authors performed strict scientific observation of their data collection. The data set is robust and is discussed adequately in the discussion. Statistical tools have been used appropriately and the data set has been rigorously and thoroughly explored.

Additional comments

The article is well written and the authors discussed objectively their findings. The use of the ABR is welcome and provide an objective assessment of the efficiency of their method. Although the data set does not suggest the chosen targets to be relevant for treating tinnitus, this work is scientifically executed and deserve its place in te litterature, as it is very informative theoretically and technically.

---

## Round 0.2 · Minor Revisions

Dear Authors, There are minor revisions needed please proceed to do this and resubmit the manuscript.Thank you for maintaining the standard of your manuscript so as to make the work reproducible and the publication citable.

·

Basic reporting

Acceptable.

Experimental design

Acceptable.

Validity of the findings

Acceptable.

Additional comments

This revised manuscript has been substantially improved. My questions have been satisfactorily answered.
The descriptive data, e.g., Fig 4, certainly appear equivalent between treatment conditions (i.e., brain stimulation) and groups (recording site). This, of course, supports the conclusions.
Suggested minor improvements.
The authors discuss the importance of recording electrode placement for ABR acquisition. Equally, or perhaps more important is consistency of sound presentation. Sound presentation features should be more precisely explained, how was the speaker presenting sound to the right ear (lines 146 – 147) arranged and standardized between recording sessions? Our experience shows that more ABR between subject variance can be attributed to sound stimulus variation than to electrode placement.
Line 176. Use “data” as a plural noun.
Line 276. Substitute “why” for “that.”
Although not that important for the present findings, the discussion paragraph considering why HF-DBS might disrupt tinnitus but not exteroceptive hearing (lines 291 -298) could be improved. In this paragraph HF-DBS is initially considered as having broad volume impact. Later in the paragraph it is hypothesized that HF-DBS might disrupt tinnitus because of a focal effect, leaving parallel pathways carrying external sound information unaffected. Overall this appears contradictory. A related consideration (maybe too distant to be included) is that most researchers now consider the pathophysiological underpinning of tinnitus to be broadly distributed in the CNS.

---

## Round 0.3 · accepted · Accept

Dear Authors,

Congratulations,Your manuscript is ready to be accepted and undergo production processing by PeerJ.